# Doxorubicin–Gelatin/Fe_3_O_4_–Alginate Dual-Layer Magnetic Nanoparticles as Targeted Anticancer Drug Delivery Vehicles

**DOI:** 10.3390/polym12081747

**Published:** 2020-08-05

**Authors:** Chiung-Hua Huang, Ting-Ju Chuang, Cherng-Jyh Ke, Chun-Hsu Yao

**Affiliations:** 1Department of Medical Laboratory Science and Biotechnology, Central Taiwan University of Science and Technology, Taichung 40601, Taiwan; chhuang@ctust.edu.tw; 2Department of Biomedical Imaging and Radiological Science, China Medical University, Taichung 40202, Taiwan; amy632825@gmail.com; 3Biomaterials Translational Research Center, China Medical University Hospital, Taichung 40202, Taiwan; 4School of Chinese Medicine, China Medical University, Taichung 40402, Taiwan; 5Department of Biomedical Informatics, Asia University, Taichung 41354, Taiwan

**Keywords:** magnetic nanoparticles, targeted drug delivery, doxorubicin, gelatin, alginate

## Abstract

In this study, magnetic nanoparticles composed of a core (doxorubicin–gelatin) and a shell layer (Fe_3_O_4_–alginate) were developed to function as targeted anticancer drug delivery vehicles. The anticancer drug doxorubicin (DOX) was selected as a model drug and embedded in the inner gelatin core to obtain high encapsulation efficiency. The advantage of the outer magnetic layer is that it targets the drug to the tumor tissue and provides controlled drug release. The physicochemical properties of doxorubicin–gelatin/Fe_3_O_4_–alginate nanoparticles (DG/FA NPs) were characterized using scanning electron microscopy, Fourier transform infrared spectroscopy (FTIR), and X-ray diffraction. The mean diameter of DG/FA NPs, which was determined using a zeta potential analyzer, was 401.8 ± 3.6 nm. The encapsulation rate was 64.6 ± 11.8%. In vitro drug release and accumulation were also studied. It was found that the release of DOX accelerated in an acidic condition. With the manipulation of an external magnetic field, DG/FA NPs efficiently targeted Michigan Cancer Foundation-7 (MCF-7) breast cancer cells and showed in the nucleus after 6 h of incubation. After 12 h of incubation, the relative fluorescence intensity reached 98.4%, and the cell viability of MCF-7 cells decreased to 52.3 ± 4.64%. Dual-layer DG/FA NPs could efficiently encapsulate and deliver DOX into MCF-7 cells to cause the death of cancer cells. The results show that DG/FA NPs have the potential for use in targeted drug delivery and cancer therapy.

## 1. Introduction

Cancer is one of the leading causes of death globally, with about 1 in 6 deaths attributed to cancer. In most chemotherapy treatments, drugs are distributed over the whole body and cause side effects on healthy tissues. Over the past three decades, research aimed at nanoparticle-based nanomedicines has increased at an incredible rate. Regarding cancer therapy, there has been particular interest in drug-loaded magnetic nanoparticles (MNPs), which provide targeted drug delivery and have the potential to reduce the adverse effects caused by chemotherapy drugs [1,2].

MNPs are made of magnetic elements, usually oxides of iron [3], cobalt [4], or nickel [5], and exhibit special properties, including small size, low toxicity, easy surface modification, and magnetic properties. Thus, MNPs serve as outstanding targeted drug delivery vehicles. Magnetic drug targeting is a method by which drugs are attached or encapsulated with MNPs and manipulated by an external magnetic field to reach the targeted area. It can optimize the drug dosage and reduce the negative effect on healthy tissues. Lee et al. demonstrated that low drug dosage (at 1/1000th of the standard dose) of MNPs can obtain excellent tumor regression [6].

Among the different kinds of MNPs, iron oxide (Fe_3_O_4_), with its ideal biocompatibility, tunable surface modification, and superparamagnetic properties, has attracted the most interest in biomedical applications [7,8]. However, uncoated Fe_3_O_4_ NPs can be easily oxidized in the air, thus reducing magnetism and colloidal stability [9]. They can also be recognized and cleared out by the reticuloendothelial system before reaching the targets [10]. Moreover, uncoated Fe_3_O_4_ NPs tend to aggregate due to strong magnetic dipole–dipole attractions [11]. Thus, a coating on the surface can not only avoid these problems but also increase the stability of Fe_3_O_4_ NPs [12,13]. Chemical compounds, such as small molecules (oleic acid, carboxylates, etc.), polymers (dextran, gelatin, polyvinyl alcohol, etc.), and inorganic materials (gold, silver, etc.), are common materials for coating [8]. Natural polymers, such as gelatin, alginate, chitosan, are ideal materials for coating MNPs.

Gelatin (gel), produced by partial hydrolysis of collagen [14], is a polyampholyte that has multifunctional groups (–NH_3_^+^, –COO^−^) accessible for various chemical modifications and able to interact with hydrophobic and hydrophilic drugs [15]. Alginate, derived primarily from brown seaweed and bacteria, is a natural anionic polysaccharide and composed of 1,4-connected β-d-mannuronic acid and α-l-guluronic acid monomers [16]. Alginate chelates with divalent cations to form hydrogel or nanoparticles and also interacts with cationic molecules through electrostatic reactions, thus making it suitable as a drug carrier [17,18]. Due to their ideal properties of low cost, nonantigenicity, and ideal degradation rate, gelatin and alginate are commonly applied in biomedical applications. For example, MNPs coated or embedded within gelatin/alginate play an essential role in developing drug delivery systems [19].

In this study, the anticancer drug doxorubicin (DOX) was selected as a model drug for delivery. DOX is an antibiotic derived from *Streptomyces peucetius* [20] and has extensive use as a chemotherapeutic drug to treat cancer, including breast cancer [21], bladder cancer [22], liver cancer [23], and childhood lymphocytic leukemia [24]. The antiproliferation and cytotoxicity effects of DOX are related to the inhibition of DNA synthesis and topoisomerase II and the increase of free radical formation and lipid peroxidation [25]. During medication, patients may experience side effects, including nausea, vomiting, diarrhea, weakness, and tiredness. It has been reported that DOX induces cardiotoxicity through the upregulation of death receptor-mediated apoptosis in cardiomyocytes [26]. Moreover, increased reactive oxygen species (ROS) are also associated with DOX-induced cardiotoxicity [27].

This study aimed to construct and characterize dual-layer DG/FA MNPs as targeted anticancer drug delivery vehicles. The embedding of DOX in the inner gelatin core prevents the risk of cardiotoxicity. The alginate outer layer provides a controlled drug release rate and stabilizes the structure of delivery vehicles. Particle characterization, drug release profile, and functional analysis of DG/FA MNPs were evaluated along with cell viability using Michigan Cancer Foundation-7 (MCF-7) cells.

## 2. Materials and Methods

### 2.1. Materials

Type A gelatin and alginate were purchased from USBiological Life Science. Ferrous ferric oxide (Fe_3_O_4_) was obtained from Liwei Nano Tech Co. LTD. Doxorubicin was purchased from LC Laboratory. Polyvinyl alcohol (PVA) and calcium chloride (CaCl_2_) were obtained from Sigma-Aldrich. Dulbecco’s modified Eagle medium (DMEM), fetal bovine serum (FBS), and trypsin-Ethylenediaminetetraacetic acid (EDTA) were purchased from Gibco Lab. MCF-7 breast cancer cell line was obtained from American Type Culture Collection (ATCC). PrestoBlue cell viability reagent was purchased from Thermo Fisher Scientific.

### 2.2. Preparation of DOX–Gel/Fe_3_O_4_–Alginate Nanoparticles (DG/FA NPs)

First, 10% (*w*/*w*) gelatin solution was prepared by dissolving 10 g of gelatin in 90 g of distilled water with continuous stirring at 50 °C. Then, DOX was dissolved in the gelatin solution and made at the concentration of 200 μM. Then, 100 μL of the DOX–gel mixture and 100 μL of 10% PVA were slowly added into 10 °C olive oil to produce the primary emulsion for the inner core components. Next, 4% (*w*/*w*) alginate solution was prepared by dissolving 4 g of alginate in 96 g of distilled water with vigorous stirring. Then, 0.5 g of Fe_3_O_4_ was added into the 4% alginate solution. After homogenization, the Fe_3_O_4_–alginate solution was slowly added into the olive oil that contained the DOX–gel nanoparticles. After that, 100 μL of 10% CaCl_2_ and 100 μL of 10% PVA were added into it, and the mixture was then homogenized with Q700 sonicator for 10 min. Finally, the magnetic DG/FA NPs were gathered by centrifugation at 1000× *g* rpm at 10 °C for 10 min.

### 2.3. Characterization of DG/FA NPs

Particle size and surface charge/zeta potential of DG/FA NPs were characterized using Zetasizer Nano (ZS90, Malvern Instruments). The mean size of the NPs was determined from the average of 100 NPs. The morphology of the DG/FA NPs was analyzed with a transmission electron microscope (TEM, HT7700, Hitachi, Tokyo, Japan). The functional groups of the NPs were characterized using Fourier transform infrared (FTIR) spectroscopy. FTIR was performed on a KBr pellet in the range of 1000–4000 cm^−1^. The crystallinity of the DG/FA NPs was studied using X-ray diffraction analysis.

### 2.4. Evaluation of DG/FA NPs Stability

The stability of DG/FA NPs was studied by measuring the change in particle size with Zetasizer Nano. After soaking in deionized H_2_O adjusted to pH of 4, 6, and 8 for various periods, the particle size of DG/FA NPs was measured. To evaluate the effect of temperature on the stability of MNPs, DG/FA NPs were soaked in distilled water at 4, 25, 37, and 42 °C for various periods, and the particle size was then measured.

### 2.5. Evaluation of Encapsulation Efficiency (EE)

Drug encapsulation efficiency was evaluated by separating the NPs from the aqueous suspension at 1000 rpm at 10 °C for 10 min. The concentration of free DOX was measured using an ELISA plate reader (Molecular Devices) at a wavelength of 470 nm. The percentage of EE of DOX—loaded NPs was calculated according to the following equation: EE (%) = (total amount of DOX—the amount of free DOX in the supernatant)/total amount of DOX. The process was performed in triplicate.

### 2.6. Evaluation of In Vitro Drug Release

The standard curve of DOX was established with an ELISA plate reader at 470 nm. DG/FA NPs were suspended in 10 mL of deionized H_2_O at different temperatures (4, 25, 37, and 42 °C). The suspension was removed at proper time intervals. The aqueous suspension was separated from NPs at 1000 rpm for 10 min and analyzed by an ELISA plate reader at 470 nm. Each test was repeated in triplicate. The release profiles were plotted as the relative percentage of DOX against time.

### 2.7. In Vitro Cytotoxicity Evaluation

The human breast cancer cell line (MCF-7) was cultured in DMEM supplemented with 10% FBS at 37 °C in a humidified atmosphere containing 5% CO_2_. The cytotoxicity effect of free DOX and DG/FA NPs against MCF-7 were measured using the PrestoBlue cell viability reagent and analyzed by an ELISA plate reader at 570 nm. MCF-7 cells were seeded in a flat-bottomed 96-well culture plate at a density of 1 × 10^4^ cells/well for 24 h. Then, various concentrations of free DOX, G/A NPs, G/FA NPs, and DG/FA NPS were added to the wells and incubated for 48 h. The cell viability was evaluated using PrestoBlue cell viability reagent. The % cell viability relative to the control was calculated according to the following formula:% cell viability = (A_570 nm_ of test cells/A_570 nm_ of control cells) × 100(1)

### 2.8. Drug Delivery and Accumulation of DOX and DOX-Loaded MNPs

MCF-7 cells were seeded in a flat-bottomed 96-well culture plate at a density of 1 × 10^4^ cells/well for 24 h, and the plasma membrane was then stained with 1 μM DiO for 15 min. After washing the plates with phosphate buffered saline (PBS) three times, the suspensions of DG/FA NPs and 4 μM of DOX were added into the wells. The cells were incubated for different time intervals and with or without the application of the external magnetic field. Before the fluorescence measurements, MCF-7 cells were washed with PBS and fixed with paraformaldehyde. The cellular uptake and accumulation of DOX were analyzed using high-content screening (Molecular Devices).

### 2.9. Statistical Analysis

The results are presented as the mean ± standard deviation. Statistical analysis was conducted using Student’s *t*-test or one-way analysis of variance followed by a post hoc Fisher’s least significant difference test for multiple comparisons. The level of statistical significance was set at *p* < 0.05.

## 3. Results and Discussion

### 3.1. Characteristics of Prepared Nanoparticles

#### 3.1.1. Particle Size, Zeta Potential, and Encapsulated Rates

Numerous techniques have been reported to fabricate Fe_3_O_4_ NPs, and the particle sizes have ranged from 9 to 800 nm [8]. The size of NPs may affect their biodistribution. Small-sized NPs possess good penetration but poor retention, while large-sized NPs have the opposite feature. Generating NPs with suitable size may be crucial to obtain proper pharmacokinetic and elimination behavior.

The particle size, zeta potential, and encapsulated rate of various nanoparticles are shown in Table 1. The mean diameters of gelatin nanoparticles (G NPs), alginate nanoparticles (A NPs), gelatin/alginate nanoparticles (G/A NPs), DOX–gelatin/alginate nanoparticles (DG/A NPs), and DOX–gelatin/Fe_3_O_4_–alginate nanoparticles (DG/FA NPs) were 318.5 ± 4.4, 366.6 ± 4.7, 325.0 ± 5.0, 370.5 ± 2.8, and 401.8 ± 3.6 nm, respectively. With the addition of DOX and Fe_3_O_4_, the diameter of NPs increased by approximately 45 and 29 nm, respectively.

The average diameter of the DG/FA dual-layer magnetic nanoparticles was 401.8 ± 3.6 nm. Biological barriers are factors in determining the biodistribution of NPs. Therefore, the efficiency of drug delivery and accumulation profiles were further analyzed. One of the mechanisms for the entry of NPs into tumors is based on the enhanced permeability and retention (EPR) effect [28,29]. In a recent research, Warren Chan and colleagues demonstrated that NPs are transported into solid tumors by endothelial cells mainly through an active process of transcytosis [30]. As the EPR effect cannot be considered as a general principle, other than the particle size, enhancement in the cellular uptake of NPs through active transcytosis is a challenge and remains to be further studied.

As can be seen in Table 1, the mean zeta potential of G NPs was 3.5 ± 0.7 mV, and this may be attributed to the amino acid components of gelatin. Gelatin polymer chains include approximately 12% negatively charged side chains (glutamate and aspartate), approximately 13% positively charged side chains (lysine and arginine), and 11% of hydrophobic side chains (leucine, isoleucine, valine, etc.) [31]. The various side chains make it able for gelatin to interact with hydrophilic and hydrophobic molecules and make gelatin a versatile drug carrier. As shown in Table 1, the encapsulated rates of DG/A NPs and DG/FA NPs were 71.5 ± 0.4% and 64.6 ± 11.8%, respectively. Based on its high drug encapsulation efficiency, gelatin can perform as an ideal drug delivery carrier.

The A NPs possessed a mean negative surface charge (−8.5 ± 0.6 mV) (Table 1). Alginates are linear unbranched polysaccharides that contain different amounts of (1→4′)-linked β-d-mannuronic acid and α-l-guluronic acid residues. The negative zeta potential of ANPs is ascribed to these anionic components [32]. The mean zeta potential of DG/FA NPs consisting of DOX–gel inner core and Fe_3_O_4_–alginate outer layer was −0.9 ± 0.8 mV. It has been reported that there is cross-linking between alginate and gelatin [19]. The positively charged gelatin core neutralized the negatively charged alginate outer layer, and the mean zeta potential of DG/FA NPs thus increased from −8.5 ± 0.6 mV to −0.9 ± 0.8 mV (Table 1).

#### 3.1.2. Transmission Electron Microscopy Analysis

TEM images (Figure 1) displayed the morphology of DG/A NPs and DG/FA NPs. Both images showed homogenized spherical particles with a solid and dense structure. With the addition of Fe_3_O_4_, the microscopy image of DA/FA NPs revealed the dual-layer structure composed of the inner core (DOX–gelatin) and the outer coating layer (Fe_3_O_4_–alginate). The mean diameters of G/A NPs and DG/FA NPs, measured using TEM, were 175 ± 4.5 and 370 ± 5.1 nm, respectively. G/A NPs and DG/FA NPs were smaller when viewed with TEM as compared to the average particle size observed with the zeta potential analyzer (Table 1). This apparent difference between the two results may be related to the dehydration of the G/A NPs and DG/FA NPs during sample preparation for TEM imaging.

#### 3.1.3. X-ray Diffraction and Fourier-Transform Infrared Spectroscopy Analysis

The crystalline characterization of the prepared NPs was analyzed using XRD. Gelatin and alginate are amorphous polymers and obtained no specific peaks in the XRD analysis (Figure 2A). The analyzed diffraction peaks of DG/FA NPs were detected at 2θ = 35.1, 56.7, 35.1, and 62.7°, which matched well with the standard diffraction peaks of Fe_3_O_4._ As can be seen from Figure 2A, the XRD pattern of DOX showed a serial peak at 2θ = 30–38°. The characteristics peaks of DOX were not detected at the same position in the DG/FA NPs. This may correspond to the amorphous nature of DOX in DG/FA NPs. The results revealed that there existed a certain degree of interaction between DOX and the composites of DG/FA NPs.

FTIR spectra of the prepared NPs are showed in Figure 2B. DG/FA NP displayed some characteristic peaks that were similar to its components. The band at 1543 cm^−1^ (N–O stretching) and 3083 cm^−1^ (N–H stretching) can be attributed to gelatin, the band at 1413 cm^−1^ (O–H stretching) and 1611 cm^−1^ (C = C stretching) can be attributed to alginate, and the band at 1745 cm^−1^ (C = O stretching) can be attributed to DOX. The result of XRD and FTIR analysis confirmed that DOX-loaded NP was successfully prepared from its components.

### 3.2. Stability of DG/FA NPs

The stability of DG/FA NPs was evaluated based on the change of particle size in different conditions, namely, storage (4 °C, pH 7), room temperature (25 °C, pH 7), human body (37 °C, pH 7), and tumor tissue (42 °C and pH < 7). After incubation at different conditions with various temperatures and pH values for 72 h, DG/FA NPs maintained in steady condition (Figure 3A,B). Although gelatin serves as useful drug delivery for hydrophilic and hydrophobic drugs and obtains high efficiency of drug loading, it has been reported that DOX-loaded gelatin NPs show cardiotoxicity. The toxicity may be caused by the degraded drug peptide conjugates [15]. To reduce the chance of cardiotoxicity, DOX–gel NPs were coated with the magnetic alginate outer layer. Based on the results of the stability assay (Figure 3), with the protection of the alginate outer layer, the particle sizes of DG/FA NPs displayed less than 10% decrease at variable conditions. This indicates that the alginate outer layer may protect DG/FA NPs from early degradation when passing through different tissues.

### 3.3. In Vitro Drug Release Assay of DG/FA NPs

Efficient nanocarriers should not only obtain excellent drug loading efficiency but should also provide drug release in a controlled manner. Based on the results of the ELISA assay, the DOX loading efficiency was 64%, which is higher than 42% for gelatin NPs [33] and similar to 70% for gold-coated magnetite NPs [34]. The method of producing DG/FA NPs obtained high drug loading efficiency.

The DOX release rates of DG/FA NPs soaked in different conditions with various temperatures and pH values for 24 h are shown in Figure 4. The results revealed that DG/FA NPs soaked at 37 °C obtained a higher release rate. The lower release rate of DG/FA NPs soaked at 4 and 25 °C may indicate the excellent stability of DG/FA NPs when stored at 4 and 25 °C. The DOX release rates of DG/FA NPs soaked in PBS buffer of pH 4, 6, and 7 at 37 °C for 24 h were 90.55 ± 2.93%, 90.25 ± 5.69%, and 37.55 ± 5.53%, respectively. The release rates of DG/FA NPs soaked in acid solution (pH 4 and 6) were significantly higher when compared to that in pH 7 solution. The lower pH decreased the negative charge on the surface of DG/FA NPs, and the electrostatic attraction between the DOX and gelatin became weak. Thus, more DOX was released from DG/FA NPs. The extracellular pH (pH_e_) values of the intralysosomal compartment and the tumor microenvironment were 4.8 and 6.5, respectively [35,36]. In comparison, the pH_e_ of healthy tissue is approximately 7.2 to 7.5 [37].

Compared with the reported NPs [38], the highest release efficiency of iron-based nanoparticles (IONP_DOX_) is 80%. The release rate of the novel DG/FA NPs developed here was 90.55 ± 2.93%, which is higher than that of IONP_DOX._

### 3.4. In Vitro Cytotoxicity Assay of Free DOX and NPs without DOX

#### 3.4.1. In Vitro Biocompatibility Assay of NPs without DOX

G/A NPs and G/FA NPs were added into MCF-7 cells in 96-well plates. After 24 h of incubation, the cell viabilities for G/A NPs and G/FA NPs were 93.3 ± 7.3% and 90.7 ± 8.7%, respectively (Figure 5). The results indicated that G/A NPs and G/FA NPs obtained excellent biocompatibility.

#### 3.4.2. Half-Maximal Inhibitory Concentration (IC50) of Free DOX

Breast cancer cells (MCF-7) was used to evaluate the efficiency of free DOX. In Figure 5, MCF-7 cells were cultured with different concentrations of free DOX for 24 h, and the IC50 of free DOX against MCF-7 cells was about 4 μM. In order to reduce the damage to healthy cells, the concentration of DOX loaded into the DG/FA NPs was 2 μM.

### 3.5. In Vitro Studies of DOX Drug Delivery and Accumulation

Doxorubicin has been found to localize mainly in the nucleus of the sensitive cell line; proteasome plays a role in translocating DOX from the cytoplasm into the nucleus [39]. In order to study the DOX drug delivery, free DOX (4 μM) and DG/FA NPs were incubated with MCF-7 cells at various time intervals. High-content screening image acquisition analysis was used to compare fluorescent drug uptake, accumulation, and distribution. The particle uptake efficiency can be explained by variations in cell membrane composition and metabolic activity of the tested cells. Higher internalization is an indicator of more accumulation of drug molecules.

In Figure 6A, DOX and plasma membrane are displayed in red and green fluorescence, respectively. Free DOX (red fluorescence) entered into the cytoplasm and started to show in the nucleus after 1 h of incubation. As the control, the relative fluorescence intensity of DOX reached 100% after 24 h of incubation (Figure 6B). DG/FA NPs did not show in the nucleus until 12 h of incubation, and the relative fluorescence intensity of DG/FA NPs without the external magnetic force reached 51.4% after 24 h of incubation. Although the drug delivery efficiency of DG/FA NPs was slower than free DOX delivery, it indicates that the MCF-7 cells could uptake DG/FG NPs. With the use of the external magnetic field, DG/FA NPs started to show in the nucleus after 6 h of incubation and reached 98.4% and 99.1% (relative fluorescence intensity) after 12 and 24 h of incubation, respectively (Figure 6B). The manipulation of the magnetic force can facilitate drug delivery, and efficient delivery may assist cellular uptake [35]. Thus, the drug uptake efficiency of DG/FA NPs was higher than the control (free DOX) at 12 h of incubation.

### 3.6. In Vitro Cytotoxicity Assay of DG/FA NPs

MCF-7 cells were treated with free DOX and DG/FA NPs at various periods for cytotoxicity analysis. When treated with 4 μM DOX, the cell viability of MCF-7 cells decreased to 83.2 ± 8.4%, 61.9 ± 3.8%, 54.3 ± 3.4%, and 48.1 ± 3.6% after 1, 6, 12, and 24 h, respectively (Figure 7). According to the results of drug accumulation (Figure 6B), the relative fluorescence intensity of DOX obtained initial accumulation at 6 h of incubation. The 6 h accumulated DOX effectively caused cell viability to decrease to 61.9 ± 3.8% (Figure 7). When the relative fluorescence intensity of DOX increased to 100% after 24 h of incubation, the cell viability decreased to 48.1 ± 3.6%.

When MCF-7 cells were incubated with DG/FA NPs with or without an external magnetic field for 24 h, the cell viability was 47.7 ± 4.6% and 70.13 ± 6.7%, respectively. Without the external magnetic field, DG/FA NPs randomly dispersed in the culture medium. Thus, less relative fluorescence intensity of DOX (53.26%) accumulated in the cell nucleus and caused less cytotoxicity to the cells. With external magnetic force, DG/FA NPs efficiently gathered around the cell surface, thus leading to high relative DOX (100%) being accumulated in the nucleus after 24 h of incubation and resulting in only 47.7 ± 4.6% cells being alive.

Although only 2 μM DOX was loaded in the DG/FA NPs, it had similar cytotoxicity as 4 μM of free DOX. The manipulation of the external field not only attracted the drug-loaded NPs close to the target tumor tissues more efficiently (Figure 6B) but the therapeutic dosage of anticancer drugs could also be cut down, thus reducing the drug’s cytotoxicity to healthy tissues by the prepared drug carriers.

## 4. Conclusions

In this study, novel DG/FA NPs composed of a core (DOX–gel) and an outer layer (Fe_3_O_4_–alginate) were developed. The gelatin core provides a higher drug encapsulation rate, and the alginate magnetic layer enhances drug delivery and also decreases the cytotoxicity of DG/FA NPs when passing through healthy tissues. Experiments showed that the proper physicochemical properties of DG/FA NPs increased the cellular uptake into tumor cells and consequently caused cell death in the presence of an external magnetic field. In in vitro cytotoxicity assay, DG/FA NPs loaded with 2 μM DOX decreased 48% cell viability of MCF-7 breast cancer cells, which was similar to the results with 4 μM free DOX. DG/FA NPs could efficiently encapsulate and deliver chemotherapeutic DOX into the nucleus of MCF-7 cells and cause death of cancer cells. The results suggest that DG/FA NPs can function as promising targeted anticancer drug delivery vehicles.

## Figures and Tables

**Figure 1 polymers-12-01747-f001:**
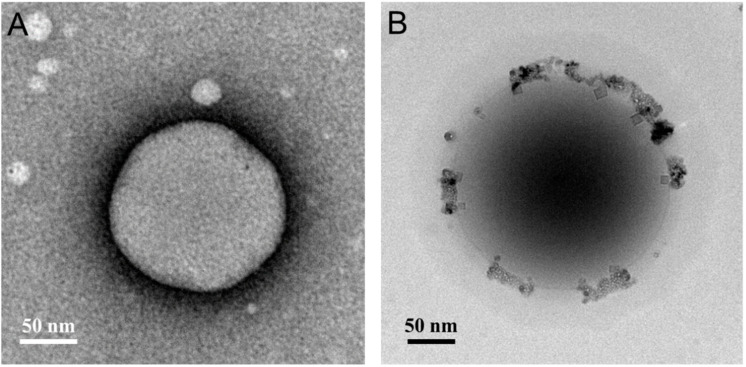
TEM characterization of NPs. (**A**) TEM image of G/A NP; (**B**) TEM image of DG/FA NP.

**Figure 2 polymers-12-01747-f002:**
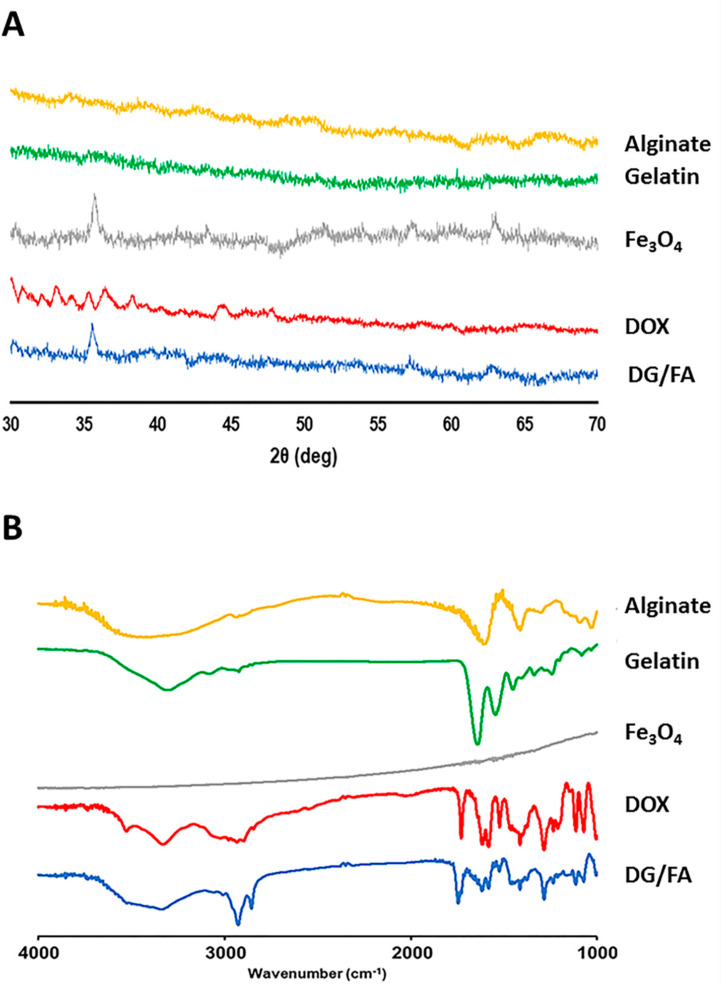
(**A**) XRD patterns of alginate, gelatin, Fe_3_O_4_ (Fe), DOX, and DG/FA NPs. (**B**) FTIR spectra of alginate, gelatin, Fe_3_O_4_ (Fe), DOX, and DG/FA NPs.

**Figure 3 polymers-12-01747-f003:**
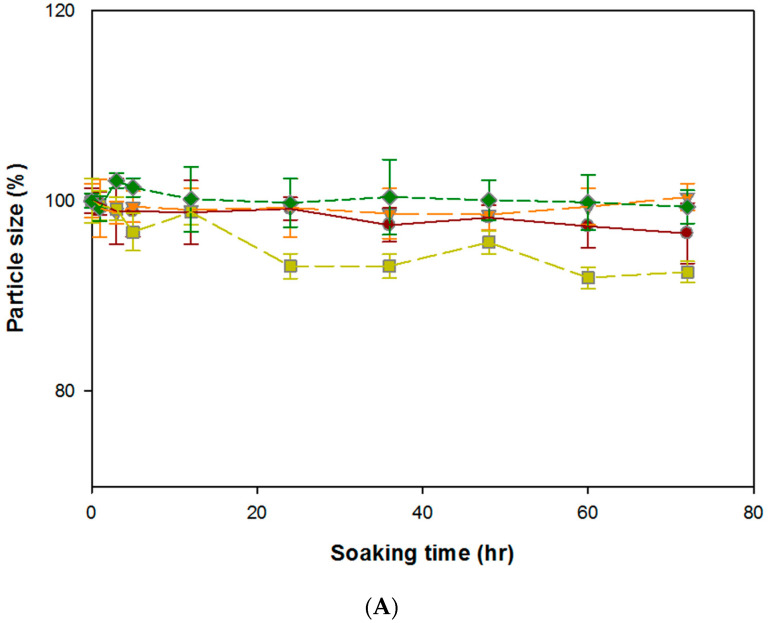
Stability characterization of DG/FA NPs. (**A**) The variation of particle size of DG/FA NPs soaked at 4, 25, 37, and 42 °C at different periods. (**B**) The variation of particle size of DG/FA NPs soaked in PBS of pH 4, 6, and 8 at different periods.

**Figure 4 polymers-12-01747-f004:**
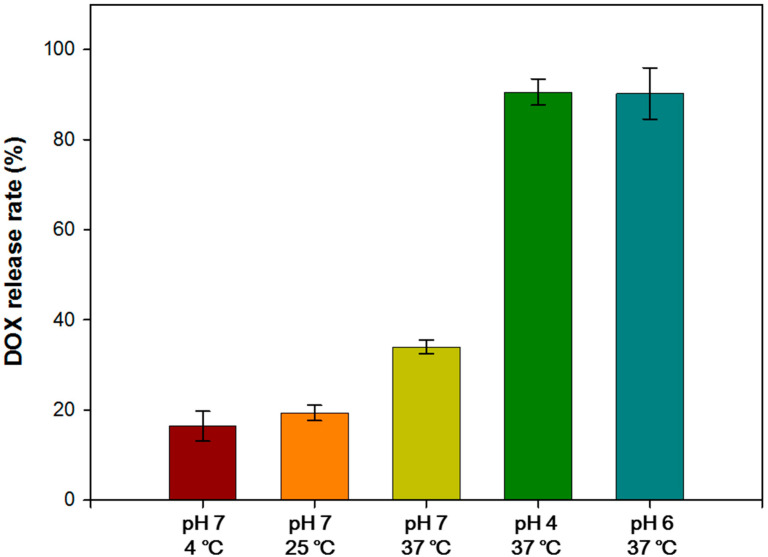
The DOX release rates of DG/FA NPs soaked in PBS of various pH values and temperatures.

**Figure 5 polymers-12-01747-f005:**
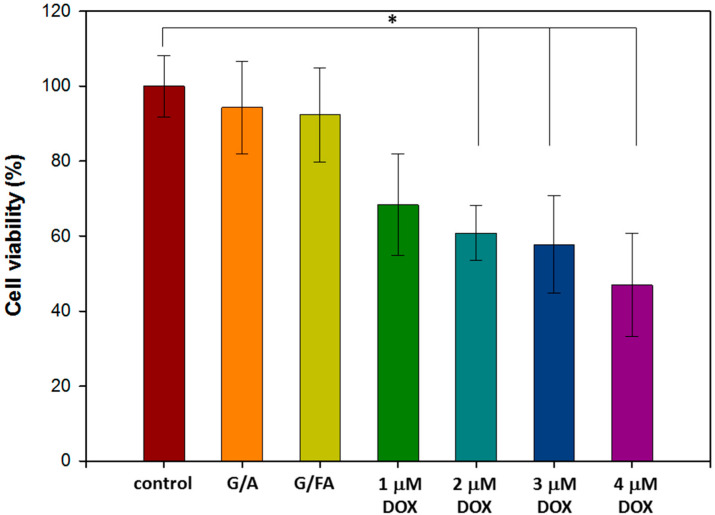
Cell viabilities of Michigan Cancer Foundation-7 (MCF-7) cells treated with G/A NPs and G/FA NPs or different concentrations of free DOX for 24 h (* *p* < 0.05).

**Figure 6 polymers-12-01747-f006:**
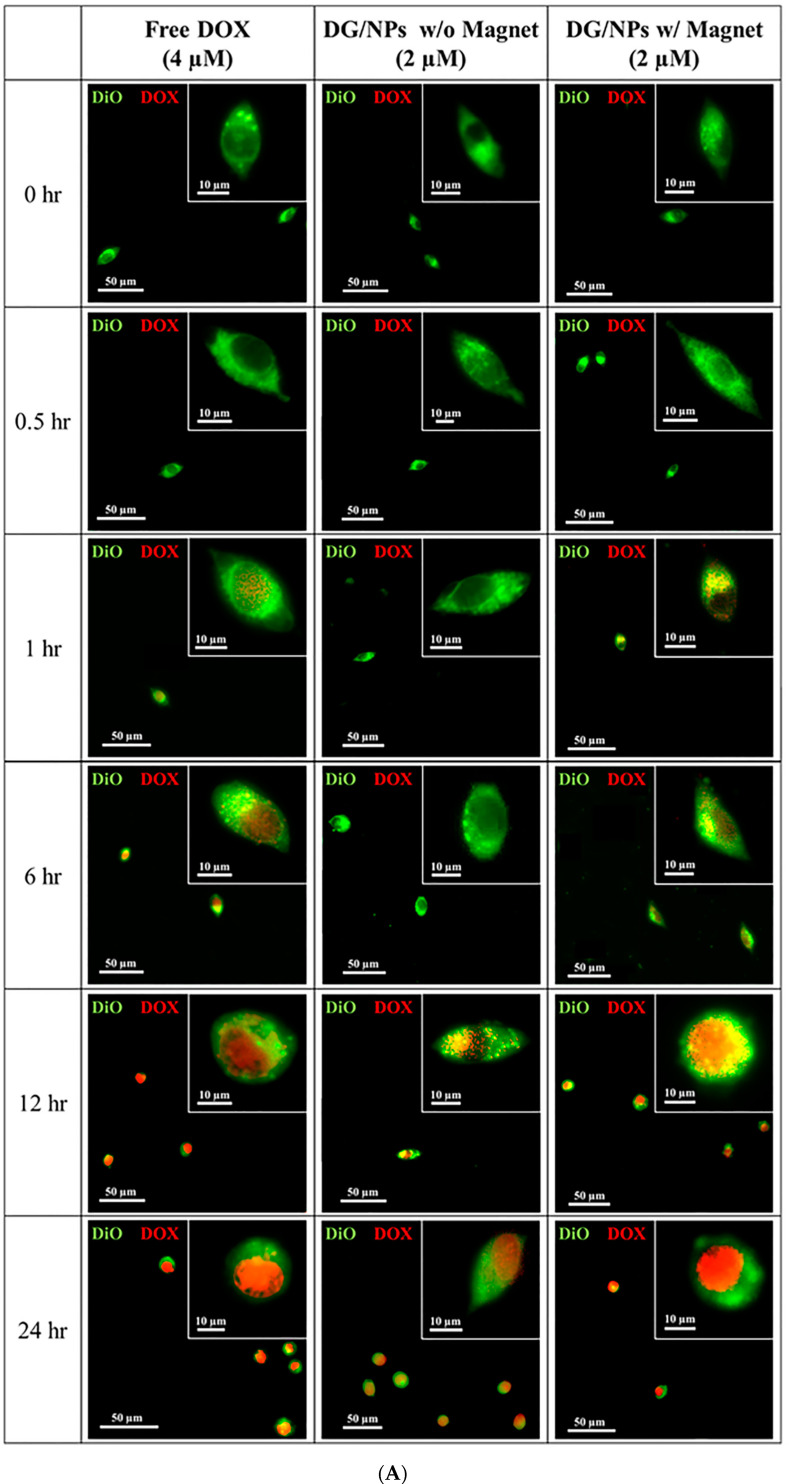
Cellular uptake and drug accumulation of MCF-7 cells treated with free DOX and DG/FA NPs with or without an external magnetic field. (**A**) Microscopy images of MCF-7 cells (DOX and plasma membrane displayed in red and green fluorescence, respectively). (**B**) DOX accumulation in MCF-7 cells at different periods (* *p* < 0.05).

**Figure 7 polymers-12-01747-f007:**
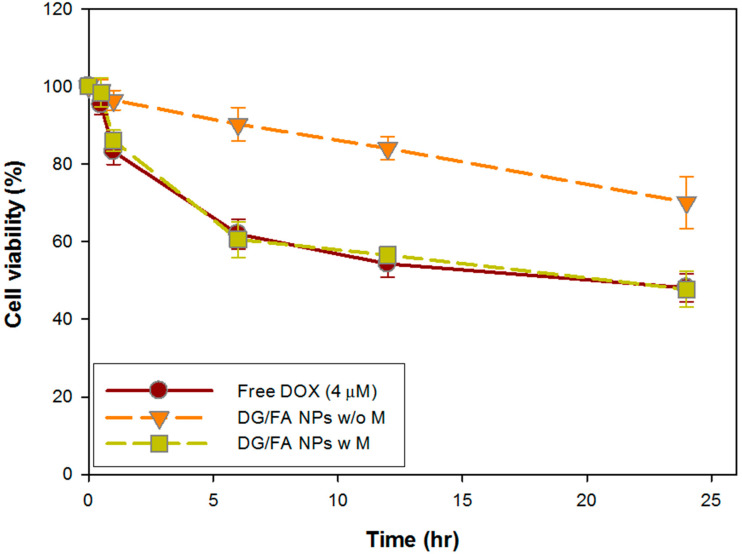
Cytotoxicity assay of free DOX and DG/FA NPs with or without external magnetic field.

**Table 1 polymers-12-01747-t001:** The particle size, zeta potential, and encapsulated rate of gelatin nanoparticles (G NPs), alginate nanoparticles (A NPs), gelatin/alginate nanoparticles (G/A NPs), doxorubicin (DOX)–gelatin/alginate nanoparticles (DG/A NPs), and DOX–gelatin/Fe_3_O_4_–alginate nanoparticles (DG/FA NPs).

	Particle Size (nm)	Zeta Potentials (mV)	Encapsulation Efficiency (%)
**G NPs**	318.5 ± 4.4	3.5 ± 0.7	-
**A NPs**	366.6 ± 4.7	−8.5 ± 0.6	-
**G/A NPs**	325.0 ± 5.0	−2.4 ± 0.5	-
**DG/A NPs**	370.5 ± 2.8	−1.5 ± 0.4	71.5 ± 0.4
**DG/FA NPs**	401.8 ± 3.6	−0.9 ± 0.8	64.6 ± 11.8

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
