# Peer review of "Doxorubicin–Gelatin/Fe_3_O_4_–Alginate Dual-Layer Magnetic Nanoparticles as Targeted Anticancer Drug Delivery Vehicles"

_polymers, 2020, doi:10.3390/polym12081747_

Round 1

Reviewer 1 Report

The manuscript entitled “Doxorubicin-Gelatin/Fe3O4-Alginate Dual-layer Magnetic Nanoparticles as Anti-Cancer Targeted Drug Delivery Vehicles” by Yao et al, describes the characterisation of doxorubicin gelatin/Fe3O4-alginate nanoparticles (DG/FA NPs), and some studies about in vitro drug release and accumulation.

Although this work is centred in an interesting aspect, as the in vitro drug release, the studies and the results are not striking at all, and the writing is rather poor. Regarding to this latter aspect, I can mention a selection of some of the problems found:

There are not keywords.

The introduction is absolutely general, and the state of the art of MNPs used as delivers for drugs is not properly described at all. In my opinion it must be fully revised.

The writing is rather sloppy (lots of mistakes, absent words, mixed fonts…), and the English needs a substantial revision.

Among many, many other mistakes I have found for instance:

L 40“The particular interest is focused…”. Who is interested?

L43 “iron, cobalt, or nickel oxide,  “ it should be oxides or are you referring to the metals…

L45 “thus play important…” Who plays?

L52 Dox embedded, should be DOX

L100 Five mg/ml??? of Fe3O4,……………………………………………………………

Experimental section: One uses a particular device for a particular thing, but one uses devices in general, … so  e.g. “ by using (singular device)” is not correct.

L112 Please, rewrite the sentence “….and treated with various pH….and temperature (4℃, 25℃, 37℃, and 42℃) respectively.

L120 DG/FA NPs were not was. What is the ELISA reader? Is it an ELISA plate reader?

L121 (4℃, 25℃, 37℃, and 42℃)  respectively, Please include respectively in the parentheses.

Although the meaning is intuitive for GNPs, ANPs, G/ANPs, DG/ANPs the first time that they are mentioned, it should be explained.

FTIR means Fourier-transform infrared spectroscopy  not Fourier transmission infrared spectroscopy

Please, pay attention to  titles:

2.2. Preparation of Dox-Gel/Fe3O4-Aligate nanoparticles (DG/FA NPs)

2.8. Drug delivery and accumulation of DOX and Dox-loaded MNPs.

  1. Results and Discussion:

Table 1 should not be a low quality image. Please, write it properly

and so on all along the text…………..

Figure 2. Pay attention to labels. There is not Fe but Fe3O4. Dox or DOX? Please, choose an only form in the manuscript.

Figure 2. (A) XRD crystallography????? of alginate. Patterns

L215 Rewrite 3.2 epigraph. E.g.: In order to evaluate the stability of DG/FA NPs in different environments????; storage (4℃), room 215 temperature (25℃), human body (37℃), and tumor tissue (42℃ and pH<7). What is the pH for the other experiments?

Fig 3. Expand de vertical axis to appreciate the behaviour to only 80-120, for instance.

In addition to all these things related to writing and editing, my mean concern is related to the absence of comparison with other “delivers” or studies related to this aspect. In fact, only four references are included in the discussion and results epigraph, and it is more a mere description of results than a thorough discussion. Consequently, my recommendation is not publish the manuscript without a major revision by the authors. At the same time author should not be so speculative to conclude “The results suggest that the DG/FA NPs could function as promising drug vehicles for targeted delivery of DOX”, as this are very simple and preliminary in vitro studies and the results are not so striking, and many other aspects have not been studied.

Author Response

The PDF file is attached.

Reviewer 2 Report

 In this paper, the authors developed a drug delivery platform composed of core (doxorubicin/gelatin) and shell layer (Fe3O4/alginate) function. They did a serial of studies to investigate the size, surface charge, drug loading, cell study, etc. Overall, the authors did lots of studies. However, the presentation of the work is poor. Please reorganize it.  Also, I have questions as follows:

  • The sentence in the line of 52-54 should give some references.
  • The logic in the introduction is not clear.  The authors give the background about the MNPs, surface decorations, dox separately, but did not give enough overview. Please reorganize the introduction.
  • In the last paragraph, it might be better to give the results about the advantages of the newly designed particle platform rather than listing what they did.
  • Why the size of the particles would increase after loading the dox? It should be embedded inside right?
  • Line 153-154 about the EPR effect, please check warren Chan’s recent papers, which prove that the EPR effect might not the main reason for permeability.
  • In line 278 ” the manipulation of the magnetic force could facilitate the DG/FA NPs to aggregate and enter into cells more efficiently”: aggregation would not facilitate the cell uptake for me.

Author Response

The PDF file is attached.

Reviewer 3 Report

The manuscript reported a method about Dual-layer Magnetic Nanoparticles for Anti-Cancer Targeted Drug Delivery Vehicles.The topic is very intresting.So I suggest the manuscript should be accepted afer minor revisions.

1.In the Fig4,Compared with the  different temperature and ,Why?What‘s the reason?

2.The authors should revise the format of Tab 1,Fig1-Fig7 in order to making it very clearly.

3.The author should revised the introuction section,Comared with the"CHEMICAL ENGINEERING JOURNAL,2020,390,124522;Chemical society reviews,2018,47(5):1874-1900;the authors  should emphasis the noverty of the manuscript.

4.Compared with the recent work,deliver efficiently is high or low?

Author Response

The PDF file is attached.

Round 2

Reviewer 1 Report

The revised version manuscript entitled “Doxorubicin-Gelatin/Fe3O4-Alginate Dual-layer Magnetic Nanoparticles as Anti-Cancer Targeted Drug Delivery Vehicles” by Yao et al. has been improved in some aspects, but it still needs to be revised as the English language (e.g. the use of articles (the, a) is totally chaotic) as well as the edition still needs some correction.

Among many other mistakes I have still found, even being formerly advised

2.2. Preparation of Dox-Gel/Fe3O4-Aligate????? nanoparticles (DG/FA NPs)

L 68  subscript and superscript for formulas

L71 -L-, instead of L-

L78 italics for Streptomyces peucetius, as well as for any other species

L108 Then 5 mg/ml of the dispersion liquid of Fe3O4 … I still miss either an amount or the concentration of the dispersion

L116  with a transmission electron microscope

L117 was performed on a KBr pellet

L240 Figure 2. (A) XRD patterns of

Figure 3. can be even more amplified, and despite amplifying the y axis, the loss of colour make the figure unclear. Please use colour again. You can also use it in other figures as 4, 5, 6 and 7 (there is no charge for that).

These are only examples, as this reviewer is not a proofreader. 

Author Response

File is attached
